# Long-Term Efficacy of Variable-Thread Tapered Implants—A Retrospective, Clinical and Radiological Evaluation

**DOI:** 10.3390/medicina56110564

**Published:** 2020-10-27

**Authors:** Oliver Blume, Eva Maria Schnödt, Michael Back, Jan IR Wildenhof, Florian A. Probst, Sven Otto

**Affiliations:** 1Maxillofacial Surgeon, Praxis Dres. Back & Blume, 80331 Munich, Germany; 2Department of Oral- and Maxillofacial Surgery, Ludwig-Maximilians-University, 80337 Munich, Germany; Florian.Probst@med.uni-muenchen.de (F.A.P.); Otto_Sven@web.de (S.O.); 3Oral Surgeon, Praxis Dres. Back & Blume, 80331 Munich, Germany; mb@backundblume.de; 4Dentist, Privatzahnklinik Schloss Schellenstein, 59939 Olsberg, Germany; jan.wildenhof@gmx.de; 5Department of Oral- and Maxillofacial Surgery, Martin-Luther-University Halle-Wittenberg, 06120 Halle (Saale), Germany

**Keywords:** dental implants, health scale for dental implants, implant success, peri-implantitis, risk factors, variable-thread tapered implants

## Abstract

*Background and Objective:* There is multifaceted evidence that variable-thread tapered implants (VTTIs) offer high primary stability but few regarding the long-term success. This retrospective clinical and radiological cohort study assessed the long-term outcome of VTTIs. *Material and Methods:* All patients treated in an OMFS practice with NobelActive Internal^®^ VTTI between October 2007 and September 2011 were invited for clinical examination. The outcome variables were (i) survival rate, (ii) implant success according to the “Health Scale for Dental Implants” and (iii) prevalence of peri-implantitis. Furthermore, the effect of local and systemic risk factors was investigated. *Results:* In 81 subjects (46 females and 35 males, mean age 65.6 years) 270 implants (157 VTTIs and 113 others as a control group) were analyzed. In 7 out of 81 patients (8.6%), 8 out of 157 VTTIs (5.1%) and 5 out of 113 other implants (4.4%) were lost. Peri-implantitis, defined as (i) presence of bleeding on gentle probing (0.25 N) or exudation and (ii) radiographic bone loss exceeding 0.5 mm since implant insertion to last follow-up, was the most common reason for implant loss (11 out of 13, 84.6%). Sixty-six out of 87 VTTIs (75.9%) were successful. Seventeen out of 42 patients (40.5%) developed peri-implantitis on 29 out of 79 VTTI sites (36.7%). Plaque and missing keratinized peri-implant mucosa were identified as potential risk factors for the development of peri-implantitis. *Conclusion:* Variable-thread tapered implants showed good long-term results, even in low bone quality. Peri-implantitis was the most common reason for implant failure and may be connected to certain risk factors.

## 1. Introduction

Many studies demonstrate long-term efficacy of rotationally symmetrical titanium implants with survival rates of up to 97% after nine years [1,2,3,4,5]. Nowadays dental implantation is frequently used and as most patients prefer fixed prostheses, the demand is constantly rising, even in patients presenting less-than-optimal conditions. The NobelActive Internal^®^ (NAI) implant system (Nobel Biocare AB, Gothenburg, Sweden) was promoted to expand the indication of implant treatment, in particular to enable implantologists to treat patients with low bone quality [6]. This variable-thread tapered implant system has apical drilling blades, a bone condensing thread design and a TiUnite^®^ surface and is used for immediate function protocols and in low bone quality. Studies show that those implants can reach high primary stability [7] and survival rates higher than 95% in the short- and medium-term [8,9].

Peri-implantitis is considered a major problem in dental implantology, as this inflammatory disease affects both peri-implant soft and hard tissue and can lead to implant failure [10]. It shares many features with periodontitis [11] but seems to progress faster. There is strong evidence that history of severe periodontitis, lack of regular follow-up and inadequate plaque-control increase the risk of peri-implantitis [12,13].

However, only few clinical studies have analyzed variable-thread tapered implants beyond implant survival. In particular, no long-term studies exceed seven years of follow-up or evaluate peri-implantitis, which is a major reason for late complications in dental implantology. Consequently, there is a lack of information concerning this implant type, which hampers the surgeons’ ability to make an informed decision about implant treatment, especially in areas of restricted indication.

Therefore, the aim of this study was to answer the following clinical questions: Do variable-thread tapered implants show satisfying long-term-results? Are there significant differences in the long-term outcomes between variable-thread tapered implants and other implants (control group), and if yes, which ones? Are there risk factors which influence the outcome?

Based on minimum success criteria of 85% survival rate after five years in mouth, the null hypothesis, “15% of variable-thread tapered implants (VTTIs) have been lost seven to eleven years after implantation”, was formed.

## 2. Material and Methods

Between 10/2007 and 09/2011, 241 patients had received a NAI (NobelActive, Nobel Biocare AB, Gothenburg, Sweden) variable-thread tapered implant (VTTI) placed by one of three surgeons of the single maxillofacial medical practice “Dres. Back und Blume”, Munich. Standard inclusion criteria for implantation were (i) the physical and mental ability to undergo surgical and prosthodontic treatment, (ii) no radiation exposure of head and neck in the last two years and (iii) no intravenous treatment of malignant tumors with bisphosphonates. Implantations were performed according to the manufacturer’s instructions. All of these 241 patients (100%) were identified and invited to our retrospective mono-centric clinical observation.

Six (2.5%) patients were deceased, 80 (33.2%) could not be reached and 74 (30.7%) declined to participate; thus, 81 (33.6%) patients were, in the end, assessed. All of those 81 subjects eligible for study inclusion were treated with at least one variable-thread tapered NAI implant in the mentioned period, were at least 18 years old at study commencement, agreed to be enrolled in the study and attended the follow-up visit. All patients gave written informed consent after being informed in detail about the objectives of the study. If there were other implants inserted until 30 September 2011, in the same patient population, those were included as a control group. For all patients, initial radiographs from the day of implant insertion or up to one week later existed. To evaluate current bone level, only existing radiographs of the implants in question and not older than two years were included in the study. The data collection of the surgical procedure and further factors such as classification of bone quality proposed by Lekholm and Zarb [14], was performed retrospectively by analyzing patient and surgery records.

All patients underwent a single follow-up visit between January 2019 and May 2019, for the purpose of this study, and were examined by one single dentist (EMS). The patients were asked to provide information on existing health conditions and systemic and local risk factors. Local risk factors as well as clinical parameters necessary for the major outcome variables were assessed. Patients were asked to rate their overall satisfaction with their implants, on a scale from 1 (very unsatisfied) to 10 (very satisfied). The radiological bone loss was calculated by comparing the initial and the most recent (no older than two years) orthopantomogram (OPG), cone beam computed tomography (CBCT) or periapical radiograph. Analogue radiographs were digitalized by camera (Canon Eos 7D, lens: 100 mm Macro, flash: Macro Ring Lite MR-14EX; Canon, Tokyo, Japan). Two-dimensional radiographs were analyzed, using CLINIVIEW version 4.22.10099 and VixWinPro version 1.5f (KaVo Dental, Biberach an der Riß, Germany), three-dimensional CBCTs (cone beam computed tomography) using iCATVision and eXam Vision version 1.9.3.13 (CT Dent Ltd., London, UK and KaVo Dental, Biberach an der Riß, Germany). In case of CBCTs, a two-dimensional picture similar to OPG was reconstructed for better comparability. Images were calibrated using implant length and evaluated by one of the authors (EMS). Bone loss was calculated by marginal bone level changes mesial and distal of implant between initial and most recent radiograph. The larger difference of both calculations mesial and distal of the implant was used as the bone loss for the implant in an approach resembling that of Derks et al. [10]. Implant survival, implant success and peri-implantitis were the evaluated outcome variables. Implant survival was assessed for all implants in mouth at the time of the follow-up.

Implant success was defined according to the “Health Scale for Dental Implants” (HSDI), as presented at The International Congress of Oral Implantologists (ICOI) Pisa Consensus Conference [15].

Peri-implantitis was defined stricter than suggested in the “2017 World Workshop on the Classification of Periodontal and Peri-Implant Diseases and Conditions” [16], specifically as (i) the presence of bleeding on gentle probing (0.25 N) or exudation and (ii) radiographic bone loss from implant placement to last follow-up exceeding 0.5 mm. Mucositis was defined as (i) the presence of bleeding on gentle probing (0.25 N) or exudation and (ii) maximum radiographic bone loss since implant placement of 0.5 mm. Peri-implant health was diagnosed when neither bleeding on gentle probing (0.25 N) nor exudation was present.

Both implant success and peri-implantitis were analyzed just for implants with recent radiographs (no older than two years).

Descriptive and inductive statistics were computed using SPSS version 26 (IBM, Armonk, NY, USA). Results are expressed as percentages or as mean values including range. Means were compared by statistical testing (Mann–Whitney Test, Pearson Chi-Square), where *p* < 0.05 was considered to be significant.

This study was approved by the local ethical committee (No. 18-787, on date: 02.01.2019.).

## 3. Results

### 3.1. Baseline Characteristics

Eighty-one patients (46 females and 35 males, mean age 65.6 years) with 270 implants (157 NobelActive Internal^®^ and 113 others: 40 Brånemark Mk III Groovy, 19 NobelSpeedy Groovy, 17 Brånemark System Mk IV, 10 NobelReplace Straight Groovy, seven Brånemark System Mk III, four Brånemark System Mk II, two Brånemark System, one NobelSpeedy Shorty (all Nobel Biocare AB, Gothenburg, Sweden), seven Astra Tech^®^ OsseoSpeed^®^ S (Dentsply Sirona, York, PA, USA), three Biomet3i^®^ OSSEOTITE (Zimmer Biomet, Warsaw, IN, USA),two Dentsply Sirona^®^ Xive (Dentsply Sirona, York, PA, USA) and one Straumann^®^ SLActive (Straumann, Basel, Switzerland)) were included in this study. Implant surfaces were 227 anodic oxidized (TiUnite^®^, Nobel Biocare AB, Gothenburg, Sweden), 30 machined (Brånemark), 10 sand blasted and acid etched (seven OsseoSpeed^®^, two Friadent^®^ plus surface and one SLActive^®^) and three acid-etched (OSSEOTITE^®^).

For 81 of 157 VTTIs (51.6%) and 49 of the other implants (45.0%), the operating surgeon decided on a submerged healing mode, while 76 VTTIs (48.4%) and 60 other implants (55.0%) healed in non-submerged. For two implants of the control group, healing mode was not documented.

Patient age range was 17.4–84.7 years (mean 57.0 ± 14.4) at implantation of the first VTTI and 26.0–92.4 years (mean 65.6 ± 14.5) at the time of the study. Mean time in situ between implantation and follow-up for surviving implants was 8.51 ± 0.70 years in the VTTI group and 12.46 ± 3.91 years in the control group. Seven patients (8.6%) did not report any of the assessed systemic risk factors. For assessed risk factors, see Table 1 and Table 2. Radiographs were available for 43 patients with 87 VTTIs and 71 other implants (including all lost implants). Implant success and peri-implant health was assessed for these cases only.

In general, patients were highly satisfied with the implant treatment, as demonstrated by the mean score of 9.54 (see Table 2). No statistically significant correlation between implant success and patient satisfaction on a scale from 1 (very unsatisfied) to 10 (very satisfied) could be detected.

Implants were mainly inserted in bone class 3 and 4 according to classification of bone quality proposed by Lekholm and Zarb [14], see Figure 1.

### 3.2. Implant Survival

In 7 out of 81 patients (8.6%), 8 out of 157 VTTIs (5.1%) and 5 out of 113 other implants (4.4%) were lost. Survival time of the lost VTTIs ranged from 1.5 to 8.5 years in the mouth. Peri-implantitis was the most common reason for implant loss (11 out of 13, 84.6%) as documented in explantation protocols. For one implant, spontaneous loss cannot be ruled out because of lacking patient memory. One other implant was explanted because of “loss of osseointegration” (documented in explantation protocol). Cumulative implant survival (see Table 3) for VTTIs was 94.9% (95% CI: 90.2–97.8%) on implant level and 95.1% on patient level as loss of three implants occurred in two patients. Survival rate for VTTI placed in the maxilla was 96.8% (4 losses out of 125) and 87.5% for VTTIs placed in the mandible (4 losses out of 32).

The null hypothesis, “15% of variable-thread tapered implants have been lost seven to eleven years after implantation”, based on Albrektssons minimum success criteria of a 85% survival rate after five years in the mouth [12], was tested by One-Sample t-Test and rejected for *n* = 157 with *p* = 0.000. No significant difference between VTTIs and implants of the control group was detected (Mann–Whitney Test: *p* = 0.800).

### 3.3. Implant Success

For implant success, only implants with recent radiographs were analyzed. Cumulative VTTI success according to HSDI (see Table 4) was 75.9% (95% CI: 65.5–84.4%). A typical example is given in Figure 2a,b. In the control group, implant success was 60.6% (95% CI: 48.3–72.0%), which was significantly lower in comparison to the VTTI group (*p* = 0.039; Pearson Chi-Square).

### 3.4. Peri-Implant Health

Peri-implantitis was defined as (i) presence of bleeding on gentle probing (0.25 N) or exudation and (ii) radiographic bone loss from implant placement exceeding 0.5 mm, and its prevalence was 40.5% on patient level (17 out of 42) and 36.7% on implant level (29 out of 79 VTTI sites) after 8.47 ± 0.67 years in situ (range: 7.42–10.50 years). Fourteen patients (33.3%) developed peri-implant mucositis at 20 out of 79 VTTI sites (25.3%) (for an example, see Figure 2c,d). Thirty of 79 VTTI sites (38.0%) in 11 patients (26.2%) showed healthy peri-implant conditions. The mean in situ time was 8.47 ± 0.67 years.

In the control group, 31 of 66 implant sites (47.0%) developed peri-implantitis after 12.75 ± 3.41 years mean time in situ.

Mean bone loss between implantation and follow-up for VTTI was 1.12 ± 1.50 mm, with 0.89 ± 1.10 mm in implants with healthy peri-implant conditions (range: 0.0–5.0 mm) and 2.06 ± 1.80 mm in implants with peri-implant mucositis or peri-implantitis (range: 0.6–7.6 mm).

Mean bone loss from implantation to last follow-up for other implants was 1.62 ± 1.68 mm (minimum, 0.0 mm; maximum, 7.4 mm).

With 11 out of 13 explantations (84.6%), peri-implantitis was the main reason for implant loss.

### 3.5. Correlation with Risk Factors

The relationship between systemic and local risk factors (see Table 1 and Table 2) and long-term outcome was further investigated. However, a logistic regression with the VTTI as the observational unit did not identify any of the evaluated variables to associate with implant survival or implant success, as none of the variables was univariate significant. Therefore, no connection between risk factors and VTTI survival and success can be stated.

Patient-level analysis with all implants, regardless of the implant type also showed no statistically significant correlation between peri-implantitis and systemic risk factors. On the implant level, however, a stepwise logistic regression analysis (forward: Wald) identified the local risk factors healing mode, keratinized mucosa and plaque on implant surface as presented in Table 5.

## 4. Discussion

The results of this study indicate that neither in clinical practice nor in research implant survival is a sufficient sole implant outcome variable. The “Health Scale for Dental Implants“ (HSDI), as suggested by Misch et al. [15], seems to be better suited to rate the actual state of the implant and health of peri-implant tissue. Furthermore, the four “Implant Quality Scale Groups” offered the possibility of a more differentiated evaluation. As implant success does not correlate with patient satisfaction, a close recall system for patients with implants beyond regular professional dental cleaning might be a matter of importance.

### 4.1. Implant Survival

In the literature, survival rates for NAI Implants are reported to range between 98.7% after one to three years [8] and 95.87% after seven years [9]. Derks et al. reported early implant loss of 1.3% and late implant loss of 2.4% of Nobel Biocare implants [17]. Friberg et al. showed high implant survival rates after eleven years even in poor local bone quality [18].

In this study, VTTIs reached a cumulative survival rate of 94.9% and therefore exceeded the 85% rate after five years in function, as demanded by Albrektsson’s success criteria [12]. Furthermore, the survival rates are in accordance with above mentioned studies, of which some have considerably higher case numbers. Even though 78.8% of the implants in this study were inserted in poor local bone quality and patients with a history of periodontitis or implant loss, the long-term survival rate was not significantly affected.

### 4.2. Implant Success

Albrektsson et al. proposed, in 1986, differentiated success criteria, and, ever since then, many other authors have followed, using parameters such as radiological bone level, healthy peri-implant soft tissue, esthetics and patient satisfaction [12,19]. In this study, implant success was defined according to the “Health Scale for Dental Implants” [15].

Using the same criteria, Francetti et al. calculated an implant success rate of 76.04% after 60 months of 56 implants in patients treated with full-arch rehabilitations supported by a combination of two tilted and two upright implants [20]. In another study with 180 immediately loaded implants with a TiUnite-surface and 96 months (8.0 years) of follow-up, Degidi et al. reported 45.0% success, 46.7% satisfactory survival, 6.1% compromised survival and 2.2% failure. However, that study excluded implants with an insertion torque below 25 Ncm and lost implants [2].

The high success rates of 75.9% in this study is in accordance with the results of Francetti et al. Higher failure rates of implants in comparison with Degidi et al. can be explained upon inclusion of lost implants and disregarding the final insertion torque.

### 4.3. Peri-Implantitis

Peri-implantitis was first defined in 1994 by Albrektsson and Isidor as “inflammatory reactions with loss of the supporting bone in the tissues surrounding a functioning implant” [21]. Even though there is no doubt about the importance of peri-implantitis as a major reason for late complications in dental implantology, in the current literature its prevalence varies vastly because of inconsistent definitions, methods and study designs [22]. The only biological parameter correlating with the prevalence of peri-implantitis seems to be Bleeding on Probing (BOP) [13]. As a consequence, in this study, peri-implantitis was defined even stricter than suggested by the “2017 World Workshop on the Classification of Periodontal and Peri-Implant Diseases and Conditions” [16], because it includes the initial remodeling immediately after implant placement. Therefore, the observed rates are difficult to compare to other studies. Publications based on the workshop classification or peri-implantitis, i.e., in which the initial remodeling is not taken into account, were for example reviewed by Zitzmann and Berglundh, who reported peri-implant mucositis in 80% of patients and 50% of implants while peri-implantitis occurred in 28% and 56–77% of patients and 12% and 43% of implants [23]. Mombelli, Müller and Cionca estimated the prevalence of peri-implantitis based on 29 publications to be 10% on the implant level and 20% on the patient level 5–10 years after implantation [24]. Derks and Tomasi performed a literature review and meta-analysis of 11 studies in which they estimated a weighted mean prevalence of peri-implant mucositis and peri-implantitis of 43% (CI: 32–54%) and 22% (CI: 14–30%), respectively [25]. Diagnosing peri-implantitis by bleeding on probing/suppuration and bone loss >0.5 mm compared to 12–24 months after prosthesis connection, Derks et al. found peri-implantitis at 40.1% of implants and in 45% of individuals [10].

Prevalence of peri-implantitis of variable-thread tapered implants in this study fits into the picture painted by the current literature. VTTIs seem to suffer from peri-implant diseases to an extent comparable to other implant types, however showed less peri-implant diseases than the control group. However, as “peri-implant tissue health can exist around implants with reduced bone support” [16], currently “healthy” implants might have experienced peri-implant disease with bone loss in the past. Consequently, during their time in function more implants may have been affected by peri-implantitis than the results of this study suggest in the first place.

As for this study, no baseline radiograph 12–24 months after insertion was available, and the initial bone remodeling after implant insertion/during healing is also included in the presented numbers regarding bone loss. Nevertheless, the documented long-term bone loss of 1.12 ± 1.50 mm for VTTIs is satisfying compared to 1.62 ± 1.68 mm in the control group.

### 4.4. Risk Factors

There is strong evidence that history of severe periodontitis, lack of regular follow-up and inadequate plaque-control increase the risk of peri-implantitis while evidence regarding poorly controlled diabetes mellitus and smoking is inconclusive [16,22]. In this study, no connection between peri-implantitis at VTTI and systemic risk factors could be found. This might be caused by the small sample size. Nevertheless, with regression analysis, correlations on the implant level could be detected between peri-implantitis at VTTI and the following risk factors: submerged healing mode, lack of keratinized mucosa around implant, plaque on implant, abutment or crown surface.

There is strong evidence that plaque is the etiological factor for peri-implant mucositis [11,16], and peri-implant mucositis is treated as a predecessor of peri-implantitis. Thereby, the 5.457-times higher chance of peri-implantitis of VTTI covered by plaque in comparison to those not covered by plaque calculated in this study can be explained.

Lately, excess cement or cement remnants were discussed as a risk factor for the development of peri-implantitis [26,27]. In this study, however, no correlation between retention mode and occurrence of peri-implantitis could be detected.

Missing keratinized mucosa around the implant may compromise patient comfort and lead to peri-implantitis because of complication of plaque removal [16]. On the other hand, peri-implant inflammation can lead to recession and loss of keratinized mucosa. The direction of causality between the missing keratinized mucosa and peri-implantitis cannot be determined based on the data collected during this study.

Submerged implant healing is usually preferred to non-submerged healing [28] because it is considered to bear less risk of peri-implant inflammation. A few studies showed a higher risk of implant failure in correlation with non-submerged healing [29,30]. Submerged implant healing is supposed to be less correlated with complications such as peri-implant bone loss and hence often preferred over non-submerged healing in patients with higher individual risk for peri-implantitis. Therefore, the result of the regression analysis indicating submerged implant healing as a risk factor should be interpreted with caution. Experienced surgeons prefer submerged healing, especially for patients with higher risk for peri-implantitis.

In summary, plaque on implant surface and missing keratinized peri-implant mucosa were identified as potential risk factors for the development of peri-implantitis. Further prospective clinical studies are required to confirm and estimate this risk.

### 4.5. Limitations

Even though OPG is not ideal for measurement of peri-implant bone loss, it has been used lately to asses radiological bone loss around dental implants [31]. For this study, the most recent radiograph (which in some cases was OPG) was analyzed, so no participant was exposed to additional radiation exposure.

Although retrospective clinical studies are of high clinical importance, they show some inherent flaws, such as limited convergence of data. The reduced number of participants limits the informative value of the results. Selection and recall bias hinder the identification of cause-effect-relationships. Therefore, precaution should be taken when aiming to generalize the outcome of this study to a bigger patient population.

The etiology of peri-implantitis is similar to the etiology of periodontitis [11]. Lately, Isola et al. highlighted on basis of nutraceutical agents and other mediators the interaction of periodontitis and coronary heart disease [32,33,34]. Such agents might be used in the future as early markers of periodontitis and peri-implantitis. In this study, no serum or salvia samples were analyzed.

As prospective, randomized studies are hard to implement because of various reasons, retrospective studies tighten the gap by providing multifaceted views and offering the possibility for clinicians to make a well-founded choice of implant type for the individual indication for their patients. Future studies will have to demonstrate the connection between certain risk factors, implant type and long-term variable-thread tapered implant survival, and success and prevalence of peri-implantitis.

## 5. Conclusions

Variable-thread tapered implants showed good long-term survival and success rates for more than seven years, even in low bone quality. Peri-implantitis did not seem to occur more often than at other implant types but demonstrated high prevalence on both the implant and patient level. Furthermore, it was the most common reason for implant failure and may be connected to certain risk factors, such as plaque on implant surface and missing keratinized peri-implant mucosa. The correlation between peri-implantitis and plaque should be considered as a reminder to shape prosthodontics according to optimum cleaning possibilities and to instruct patients on oral hygiene on a regular basis. Other detected correlations need to be investigated further.

As the variable-thread tapered implants in this study were often inserted in bone class 3 or 4, the conclusion arises that variable-thread tapered implants are applicable even in qualitatively compromised bone and might offer a possibility to broaden the field of indication. Further studies will have to confirm if this is true and whether it is a consequence of the high primary stability of this implant type.

## Figures and Tables

**Figure 1 medicina-56-00564-f001:**
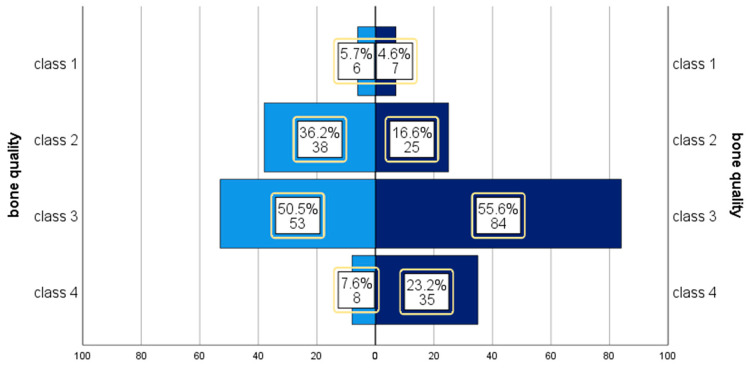
Bone quality and implant type. Left: control group. Right: variable-thread tapered implant (VTTI) group bone quality, according to Lekholm and Zarb [14].

**Figure 2 medicina-56-00564-f002:**
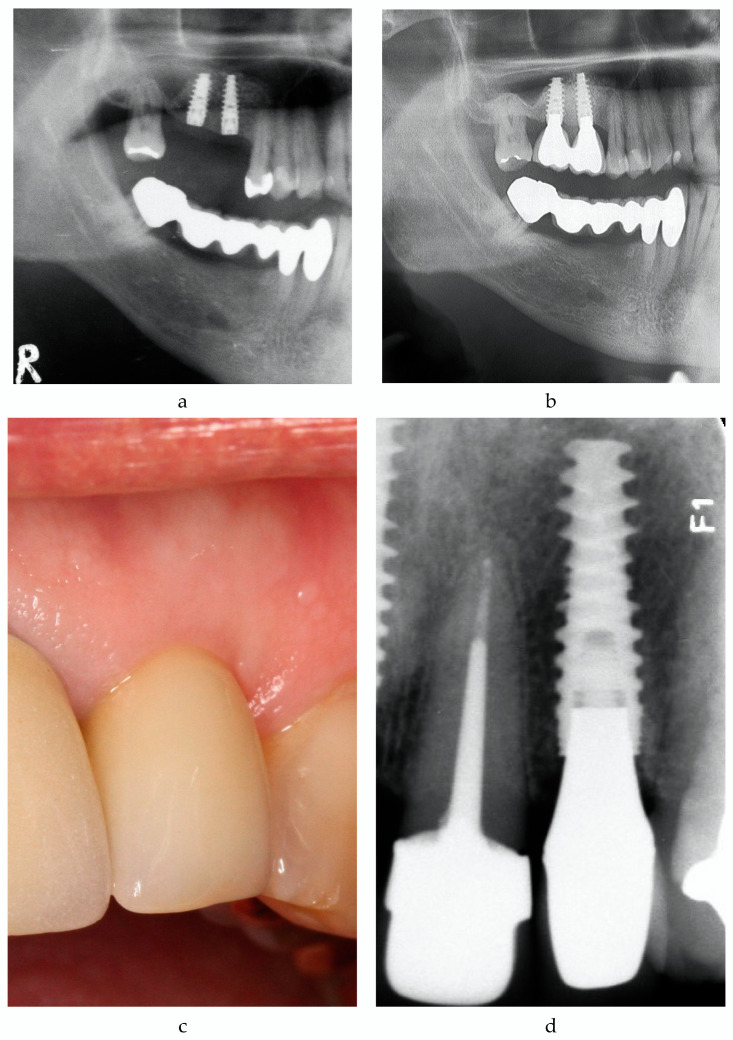
Implant success. (**a**) Patient 54 excerpt of orthopantomogram (OPG) 2008. Situation after implantation regio 16 and 17. (**b**) Patient 54 excerpt of OPG 2019; 10.5 years after implantation regio 16 and 17. NobelActive Internal^®^ (NAI) implants 16 and 17 successful (“Health Scale for Dental Implants” (HSDI) group I). (**c**) Patient 53, photograph 2019. NAI implant 9.1 years after implantation regio 22. (**d**) Patient 53, radiograph 2019. NAI implant 9.1 years after implantation regio 22, mild mucositis. Note: Figure 2c should be printed in color.

**Table 1 medicina-56-00564-t001:** Systemic and local risk factors at patient level (*n* = 81).

Risk Factor		Events	%
current smoker		6	7.4
smoking history in pack years	1–9 10–19 20–39 >40	15 6 7 7	18.5 7.4 8.6 8.6
consume of alcohol	occasionally daily	29 21	35.8 25.9
diabetes mellitus		7	8.6
HbA_1C_ in percent (*n* = 5)	<6.5 6.5–7.5 >7.5	2 1 2	40.0 20.0 40.0
history of immunosuppressant		4	4.9
Carcinosis		9	11.1
Bisphosphonate-intake at follow-up	oral intravenous	3 3	3.7 3.7
periodontitis (*n* = 80) according to PSR (Periodontal Screening and Recording)		35	43.8
oral hygiene status	insufficient	8	9.9
Bruxism		38	46.9
frequency of professional dental cleaning	never 1/year >1/year >2/year >3/year	3 14 41 16 7	3.7 17.3 50.6 19.8 8.6

**Table 2 medicina-56-00564-t002:** Results of clinical and radiological examination, including local risk factors.

	VTTI (*n* = 149)	Control Group (*n =* 108)
plaque ^1,2^	35.6% (53)	42.5% (45)
spontaneous bleeding ^2^	2.0% (3)	0.9% (1)
exudation ^2^	0.7% (1)	0.0% (0)
BOP	66.4% (99)	63.0% (68)
probing depth ^2^ in mm *	3.6 ± 1.5	3.6 ± 1.4
recession in mm ± SD	0.33 ± 0.88	1.17 ± 1.24
keratinized mucosa ^2^ in mm *	2.5 ± 2.1	1.2 ± 1.5
prosthodontics		
crown	55.0% (82)	50.9% (55)
implant-supported fixed prosthodontics	40.9% (61)	44.4% (48)
anchorage for removable prosthodontics	4.0% (6)	4.6% (5)
fixation		
cemented	74.5% (111)	66.7% (72)
screw-retained	25.5% (38)	33.3% (36)
patient satisfaction *	9.54 ± 0.941	9.30 ± 1.665
foreign body sensation	3.18% (5)	0.00% (0)
radiological bone loss in mm ^3,^*	1.12 ± 1.50	1.62 ± 1.68

* Mean ± standard deviation. ^1^ plaque at ≥1/6 of marginal implant surface, ^2^
*n*_control group_=106, ^3^
*n*_VTTI_ = 79, *n*_control group_ = 66. BOP = Bleeding on Probing.

**Table 3 medicina-56-00564-t003:** Summary: result of main objects on implant level.

Outcome Variable	*n*	Events	95.0% CI	Time In Situ *
Implant survival				
VTTI	157	149 (94.9%)	90.2–97.8%	8.51 ± 0.70
control group	113	108 (95.6%)	90.0–98.5%	12.46 ± 3.91
Implant success				
VTTI	87	66 (75.9%)	65.5–84.4%	8.55 ± 0.73
control group	71	43 (60.6%)	48.3–72.0%	12.84 ± 3.38
Peri-implantitis				
VTTI	79	29 (36.7%)	27.3–48.0%	8.47 ± 0.67
control group	66	31 (47.0%)	35.6–58.8%	12.75 ± 3.41

* Mean ± standard deviation of surviving implants (years).

**Table 4 medicina-56-00564-t004:** Implant success according to Health Scale for Dental Implants *.

Implant Quality Scale Group and Criteria	VTTI (*n* = 87)	Control Group (*n* = 71)
**I. Success (optimum health)**	66 (75.9%)	43 (60.6%)
(a) No pain or tenderness upon function		
(b) 0 mobility		
(c) <2 mm radiographic bone loss from initial surgery		
(d) No exudates history		
**II. Satisfactory survival**	2 (2.3%)	12 (16.9%)
(a) No pain on function		
(b) 0 mobility		
(c) 2–4 mm radiographic bone loss		
(d) No exudates history		
**III. Compromised survival**	9 (10.3%)	9 (12.7%)
(a) May have sensitivity on function		
(b) No mobility		
(c) Radiographic bone loss > 4 mm (less than 1/2 of implant body)		
(d) Probing depth > 7 mm		
(e) May have exudates history		
**IV. Failure (absolute or clinical failure)**	10 (11.5%)	7 (9.9%)
Any of following:		
(a) Pain on function		
(b) Mobility		
(c) Radiographic bone loss > 1/2 length of implant	1 (1.1%)	
(d) Uncontrolled exudate	1 (1.1%)	
(e) No longer in mouth	8 (9.2%)	5 (7.0%)
or scheduled for explantation		2 (2.8%)
time in situ ** (years)	8.55 ± 0.73	12.84 ± 3.38
minimum–maximum	7.42–11.29	8.22–22.02

* International Congress of Oral Implantologists, Pisa, Italy, Consensus Conference, 2007. ** Mean ± standard deviation.

**Table 5 medicina-56-00564-t005:** stepwise logistical regression analysis (forward: Wald).

	Step	Regression-Coefficient B	Standard Deviation	Wald	df	*p*	Exp (B) = OR 95.0% CI
(1)	presence of plaque	1.586	0.502	9.992	1	0.002	4.886 (1.827–13.064)
	constant	0.488	0.251	3.779	1	0.052	1.629
(2)	presence of plaque	1.568	0.527	8.863	1	0.003	4.798 (1.709–13.473)
	submerged healing mode	1.402	0.572	5.998	1	0.014	4.063 (1.323–12.478)
	constant	0.695	0.285	5.942	1	0.015	2.004
(3)	absence of keratinized mucosa	1.502	0.637	**5.553**	1	0.018	**4.489** **(1.288–15.649)**
	presence of plaque	1.697	0.560	**9.170**	1	0.002	**5.457** **(1.820–16.367)**
	submerged healing mode	1.442	0.605	**5.679**	1	0.017	**4.230** **(1.292–13.851)**
	constant	0.339	0.331	1.051	1	0.305	1.404

Values in bold have significant correlations with *p* < 0.05. df = degrees of freedom. OR = odds ratio.

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
