# Peer review of "Long-Term Efficacy of Variable-Thread Tapered Implants—A Retrospective, Clinical and Radiological Evaluation"

_medicina, 2020, doi:10.3390/medicina56110564_

Round 1
Reviewer 1 Report
The results presented in the article are relevant and of clinical interest. However, some parts of the article should be confirmed and complemented:
- Page 2, line 81 - " The radiological bone loss was calculated by comparing the initial and the most recent radiograph, either orthopantomogram (OPG), cone beam computed tomography (CBCT) or periapical…” Is OPG a reliable method to assess marginal bone loss? Has any current available literature implement this method to assess marginal bone loss?
- The tables should be numbered following their number of appearance. (Table 3 appears before Table 1&2.)
- In table 4, the number of implant success in control group “43” was misplaced.
- Page 7, line 170-185 - The article lacks a detailed statistical analysis of the results on “prevalence of peri-implantitis” and “mean bone loss”.
Author Response
Answer to Reviewer 1
The results presented in the article are relevant and of clinical interest. However, some parts of the article should be confirmed and complemented:
- Page 2, line 81 - " The radiological bone loss was calculated by comparing the initial and the most recent radiograph, either orthopantomogram (OPG), cone beam computed tomography (CBCT) or periapical…” Is OPG a reliable method to assess marginal bone loss? Has any current available literature implement this method to assess marginal bone loss?
Answer: Following passage has been added in “4.5. Limitations”:
“Even though OPG is not ideal for measurement of peri-implant bone loss, it has been used lately to asses radiological bone loss around dental implants [31]. For this study, the most recent radiograph (which in some cases was OPG) was analyzed so no participant was exposed to additional radiation exposure.”
- The tables should be numbered following their number of appearance. (Table 3 appears before Table 1&2.)
Answer: Numbering of tables has been corrected.
- In table 4, the number of implant success in control group “43” was misplaced.
Answer: Has been corrected.
- Page 7, line 170-185 - The article lacks a detailed statistical analysis of the results on “prevalence of peri-implantitis” and “mean bone loss”.
Answer:
- Prevalence of peri-implantitis is described both on patient an implant level. Which further information do you wish for?
- Further information regarding minimum and maximum bone loss in control group was added. Mean bone loss for both groups was already in the article:
“Mean bone loss between implantation and follow-up for VTTI was 1.12 ± 1.50 mm, with 0.89 ± 1.10 mm in implants with healthy peri-implant conditions (minimum: 0.0 mm, maximum 5.0 mm) and 2.06 ± 1.80 mm in implants with peri-implant mucositis or peri-implantitis (minimum: 0.6 mm, maximum 7.6 mm).
Mean bone loss between implantation and follow-up for other implants was 1.62 ± 1.68 mm (minimum: 0.0 mm, maximum 7.4 mm).”
Reviewer 2 Report
The retrospective study by Blume et al. assessed the long-term success of Variable-Thread Tapered Implants, basing the evaluation on clinical and radiological parameters. The study is well designed and it provides interesting data, since the demand of fixed prostheses, as authors said themselves, is steadily increasing.
I have some suggestion which may improve this article:
- In the Abstract authors did not mention in the "Material and Methods" that also a control group was included in the research. Please mention
- In the "Introduction" section authors should expand the description of NobelActive Internal implant system (characteristic, usage..).
- Tables and figures should be reorganized. Line 97: authors presented Table 3 before Table 1 and 2. In addition, Table 3 should be presented in Results section. In general, figures and tables should be presented in the paper after being mentioned in the main text.
- Line 76: "..follow up-visit between 01/2019 and 05/2019.
- Line 80: what is meant by "patients satisfaction"? Please clarify.
- Line 146: except peri-implantitis, which are the others reasons for implant loss?
- Line 148: "Two patients lost one implant, two further patients lost three implants each", to what is referred this sentence?
- Line 214: "Furthermore, the survival rates are in accordance with many other studies..", please provide references.
Author Response
Answer to Reviewer 2
The retrospective study by Blume et al. assessed the long-term success of Variable-Thread Tapered Implants, basing the evaluation on clinical and radiological parameters. The study is well designed and it provides interesting data, since the demand of fixed prostheses, as authors said themselves, is steadily increasing.
I have some suggestion which may improve this article:
- In the Abstract authors did not mention in the "Material and Methods" that also a control group was included in the research. Please mention
Answer: was added:
„In 81 subjects (46 female, 35 male, mean age 65.6 years) 270 implants (157 VTTI, 113 others as a control group) were analyzed.”
- In the "Introduction" section authors should expand the description of NobelActive Internal implant system (characteristic, usage..).
Answer: was added:
“This variable-thread tapered implant system has apical drilling blades, a bone condensing thread design and a TiUnite® surface and is used for immediate function protocols and in low bone quality.”
- Tables and figures should be reorganized. Line 97: authors presented Table 3 before Table 1 and 2. In addition, Table 3 should be presented in Results section. In general, figures and tables should be presented in the paper after being mentioned in the main text.
Answer:
Order of tables has been corrected. Table was moved to results section.
- Line 76: "..follow up-visit between 01/2019 and 05/2019.
Answer: has been corrected.
- Line 80: what is meant by "patients satisfaction"? Please clarify.
Answer: Has been added:
“Patients were asked for their overall satisfaction with their implants on a scale from 1 (very unsatisfied) to 10 (very satisfied).”
- Line 146: except peri-implantitis, which are the others reasons for implant loss?
Answer: The following passage has been added:
“For one implant, spontaneous loss can’t be ruled out because of lacking patient memory. One other implant was explanted because of “loss of osseointegration” (documented in explantation protocol).”
- Line 148: "Two patients lost one implant, two further patients lost three implants each", to what is referred this sentence?
Answer: Has been changed to:
“Cumulative implant survival (see Table 3) for VTTI was 94.9% (95%-CI: 90.2-97.8%) on implant level and 95.1% on patient level as loss of three implants occurred in two patients.”
- Line 214: "Furthermore, the survival rates are in accordance with many other studies..", please provide references.
Answer: Has been changed to:
“In literature survival rates for NAI Implants are reported between 98.7% after one to three [8] and 95.87% after seven years [9]. Derks et al. reported early implant loss of 1.3% and late implant loss of 2.4% of Nobel Biocare implants [15]. Friberg et al. showed high implant survival rates after eleven years even in poor local bone quality [16].
In this study, VTTI reached a cumulative survival rate of 94.9% and therefore exceeded the 85% after five years in function as demanded by Albrektsson [14]. Furthermore the survival rates are in accordance with above mentioned studies, of which some have considerably higher case numbers.”
Reviewer 3 Report
The tables are in the wrong order.
There is a problem with the N counting of each group in the table.
The reasons for the patients who were not followed up in the description of the inclusion criteria appear to be unnecessary in the article description.
Compared to VTTI implants, the composition of the other implants group consists of so many different implants.
Headings are omitted in the risk factors section of the discussion.
Author Response
Answer to Reviewer 3
- The tables are in the wrong order.
Answer:
Order of tables has been corrected.
- There is a problem with the N counting of each group in the table.
Answer:
N counting differed between tables because
- some tables refer to patients, others to implants
- some implants couldn’t be included, e.g. as radiographs were not available for all implants, peri-implantitis was analyzed just for implants with recent radiographs. This is mentioned in the text.
- The reasons for the patients who were not followed up in the description of the inclusion criteria appear to be unnecessary in the article description.
Answer:
As we thought it might be interesting for readers, we decided to mention why so many of the originally 241 patients did not participate in the study.
- Compared to VTTI implants, the composition of the other implants group consists of so many different implants.
Answer:
In this study, older implants in the same patients served as a control group. Patients were treated by various practitioners before attending to this study. Some of the implants in the control group were inserted more than 23 years before study inception; some patients received various implants by different dentists over the years.
- Headings are omitted in the risk factors section of the discussion.
Answer:
Heading “4.4. Risk factors” was inserted and “4.4. Limitations” changed to “4.5. Limitations”